# Wild Boar Effects on Fungal Abundance and Guilds from Sporocarp Sampling in a Boreal Forest Ecosystem

**DOI:** 10.3390/ani12192521

**Published:** 2022-09-21

**Authors:** Antonio J. Carpio, Marta García, Lars Hillström, Mikael Lönn, Joao Carvalho, Pelayo Acevedo, C. Guillermo Bueno

**Affiliations:** 1Grupo de Sanidad y Biotecnología (SaBio), Instituto de Investigación en Recursos Cinegéticos, (IREC UCLM-CSIC-JCCM), 13071 Ciudad Real, Spain; 2Faculty of Biological Sciences, Complutense University of Madrid, 28040 Madrid, Spain; 3Department of Electrical Engineering, Mathematics and Science, Faculty of Engineering and Sustainable Development, University of Gävle, 801 76 Gävle, Sweden; 4Department of Biology & CESAM, University of Aveiro, 3810-193 Aveiro, Portugal; 5Department of Botany, Institute of Ecology and Earth Sciences, University of Tartu, 50409 Tartu, Estonia

**Keywords:** disturbance, ectomycorrhiza, fungal guilds, rooting, *Sus scrofa*, Sweden, wild boar

## Abstract

**Simple Summary:**

Native wild boar populations are expanding across Europe, causing concern due to their significant soil disturbances and considerable impact on ecosystems. However, little is known about how wild boar activities affect other organisms. This study investigated the effects of wild boars on the abundance of fungal sporocarps and their respective fungal guilds (i.e., symbiotic, saprophytic and pathogenic) in boreal forests in Sweden. We selected 11 forested sites in central Sweden: six with and five without the presence of wild boar. We determined the presence or absence of wild boar and rooting intensity at each site. Simultaneously, we investigated the abundance of fungal sporocarps and their fungal guilds. We found that the presence of wild boar and rooting intensity were associated with the abundance of fungal sporocarps. Wild boar rooting was positively correlated with saprotrophic fungi and negatively with symbiotic fungi. Pathogenic fungi were more abundant in plots with no rooting but in the presence of wild boar. We conclude that wild boar represents a recurrent disturbance agent and, based on sporocarp abundance, could eventually affect entire fungal populations.

**Abstract:**

Native wild boar (*Sus scrofa*) populations are expanding across Europe. This is cause for concern in some areas where overabundant populations impact natural ecosystems and adjacent agronomic systems. To better manage the potential for impacts, managers require more information about how the species may affect other organisms. For example, information regarding the effect of wild boar on soil fungi for management application is lacking. Soil fungi play a fundamental role in ecosystems, driving essential ecological functions; acting as mycorrhizal symbionts, sustaining plant nutrition and providing defense; as saprotrophs, regulating the organic matter decomposition; or as plant pathogens, regulating plant fitness and survival. During autumn (Sep–Nov) 2018, we investigated the effects of wild boar (presence/absence and rooting intensity) on the abundance (number of individuals) of fungal sporocarps and their functional guilds (symbiotic, saprotrophic and pathogenic). We selected eleven forested sites (400–500 × 150–200 m) in central Sweden; six with and five without the presence of wild boar. Within each forest, we selected one transect (200 m long), and five plots (2 × 2 m each) for sites without wild boar, and ten plots for sites with boars (five within and five outside wild boar disturbances), to determine the relationship between the intensity of rooting and the abundance of sporocarps for three fungal guilds. We found that the presence of wild boar and rooting intensity were associated with the abundance of sporocarps. Interestingly, this relationship varied depending on the fungal guild analyzed, where wild boar rooting had a positive correlation with saprophytic sporocarps and a negative correlation with symbiotic sporocarps. Pathogenic fungi, in turn, were more abundant in undisturbed plots (no rooting) but located in areas with the presence of wild boar. Our results indicate that wild boar activities can potentially regulate the abundance of fungal sporocarps, with different impacts on fungal guilds. Therefore, wild boar can affect many essential ecosystem functions driven by soil fungi in boreal forests, such as positive effects on energy rotation and in creating mineral availability to plants, which could lead to increased diversity of plants in boreal forests.

## 1. Introduction

Wild boar (*Sus scrofa*) have been expanding throughout their native range and beyond during the last decades, including a recent recolonization of northern European countries such as Norway [1]. Overabundant and expanding wild boar populations increase human–wildlife conflicts (HWCs) [1]. These HWCs include traffic accidents [2], crop damage [3], disease transmission to livestock [4,5,6] and threats to sensitive areas and species [7,8,9,10,11]. 

Because wild boar can profoundly affect ecosystems, altering the environmental conditions for other organisms, the species has been described as an ecosystem engineer [12]. Thus, wild boar have a strong potential to affect ecosystem functions, particularly those based on soil processes and organisms. The effects of wild boar activities on the soil processes and biota have rarely been studied [13,14]. 

Wild boar affect forest ecosystems through rooting feeding behaviors (i.e., soil disturbance, bioturbation), as they turn over the soil in search of underground food resources [15,16]. Wild boar rooting, depending on intensity, may reduce plant cover and diversity [17], thus modifying the vegetation dynamic and regeneration at local and even regional scales [7,12]. The extent and intensity of wild boar rooting can also affect fundamental soil properties, affecting the species composition of plants and other essential organisms (i.e., soil fungi), along with their abundance and ecosystem functions [18,19,20,21]. 

Soil fungi play a fundamental role in forest ecosystems and drive several key ecological functions, such as sustaining plant nutrition and defense by mutualistic symbiosis, regulating plant fitness and survival through fungal pathogens or symbionts, and decomposition of organic matter by fungal saprotrophs [22,23,24]. The reduction in plant cover and changes in soil properties (e.g., soil moisture and soil nutrient concentration) caused by rooting could favor the persistence and presence of some fungal guilds in relation to others. 

Rodriguez-Ramos et al. (2020) [25] reported that soil fungi are particularly sensitive to disturbances (e.g., wildfire, clear-cut logging or salvage-logging) and showed that soil fungal community dynamics were altered in North American boreal forests. They also suggested that disturbance of the forest floor may promote changes in the soil fungal community composition, suggesting that soil fungi can occupy different niches relative to the disturbance frequency [26]. Bioturbation caused by wild boar rooting could physically damage mycorrhizal roots [27,28,29,30] and alter soil compaction, influencing fungal vulnerability [31,32]. Still, the indirect effect of rooting regarding mineralization may benefit fungal decomposition soil processes. When rooting, wild boar can also act as an agent of dispersion for diaspores of plants [33,34,35], fungi [36] or both [37,38]. The relative importance of wild boar as a disperser could help some fungal guilds to be promoted and therefore affect plant–fungi interactions, fungal diversity [39] and, ultimately, the dynamics of the forest ecosystems [40]. However, the net effect of the aforementioned factors on the ecosystem will most probably depend on the frequency or intensity of the disturbance [41].

Overall, the wild boar has the potential to influence fungal communities [37,38,42], either by their rooting behavior or by consuming (and dispersing) the fruiting bodies [12,43]. However, little is known about this species’ overall effects on fungi and different functional fungal guilds. Our study aimed to investigate how the presence of wild boar and the frequency of wild boar rooting (rooting intensity) affect the fungal sporocarp abundance, along with the effects on their composition in terms of fungal guilds, key actors in the boreal forest functioning. Our specific questions were: (1) does the presence of wild boar, and the rooting intensity, alter the abundance of fungal sporocarps? and (2) how influential is wild boar rooting relative to other key environmental factors (forest type, soil humidity and rockiness) regarding regulating the abundance of fungal sporocarps and their corresponding fungal guilds (symbiotic, saprophytic or pathogenic)? We hypothesize that wild boar rooting modifies the abundance of fungi, and these effects could differ depending on their functional guild. Besides the impact of soil disturbances on soil microbes, we considered other habitat factors that strongly influence the abundance of soil fungi [44,45,46,47]. Focused on the boreal forest, we have considered the degree of forest cover (openness), soil humidity and soil rockiness (which can modify humidity). We expect that environmental factors positively affect the abundance of fungi in wet and soft soils and areas with semi-open forests. 

## 2. Methods

### 2.1. Study Area

Our study area was located in the Swedish counties of Gävleborg, Uppsala and Västmanland (60°12′ N–60°37′ N, 16°36′ E–17°36′ E, WGS84; Figure 1). We selected sampling sites (independent patches of forest, at least 3 km apart from each other, in a fragmented landscape) in a boreal forest landscape that consists of natural and planted stands of Norway spruce (*Picea abies*) and Scots pine (*Pinus sylvestris*). These boreal forests also contain various species of deciduous trees, such as birch (*Betula* spp.), European aspen (*Populus tremula*) and, to a lesser extent, European oak (*Quercus robur*), common alder (*Alnus glutinosa*), common ash (*Fraxinus excelsior*), Scots elm (*Ulmus glabra*), small-leaved lime (*Tilia cordata*) and common hazel (*Corylus avellana*). These forests are, to some extent, managed forests harvested every 80 years by different types of clear cutting [48]. Mushroom picking is allowed but uncommon in these sites, and wild boar hunting is also practiced in the whole region. Some of these sites were selected with experience from an earlier study on wild boar [11], where we knew wild boar were present, but other sites were searched in the region for both the presence and absence of wild boar rooting.

The climate is representative of European boreal temperate forests with relatively cold winters and warm summers (with an average annual temperature of 6.5 °C (range −7 to 21 °C) and an annual average precipitation of 620 mm, predominantly at the end of summer. The topography of the area is relatively flat, with hills no higher than 150 m above sea level. We selected eleven sampling sites (1–1.5 ha) based on wild boar hunting yields recorded by hunters in the region (data reported to the Swedish Hunters Association). Based on the available hunting data for the 2017/2018 season [49], the sites with and without wild boar were selected in a gradient of hunted animals between 0.1 and 12.4 wild boar harvested per 1000 ha. Supplemental feeding for wild boar is allowed in Sweden [50], but there were no feeding stations close to our sites. Wild boar density (number of harvested wild boar/1000 ha) is correlated with rooting intensity on the ground, with the highest boar densities normally co-occurring with areas hosting the largest extent and number of wild boar disturbances [51]. Therefore, we used rooting intensity as an approximation to the density of wild boar at the plot level. Thus, we used rooting observed in the field to indicate the presence of wild boar and the area disturbed by boars as a proxy for wild boar density. 

### 2.2. Monitoring Design

We sampled between mid-September and mid-November 2018, coinciding with the peak of maximum sporocarp abundance [52] until the snow arrives in November. The specimens were identified through photographs, which were sent to experts. The identification was carried out mainly through the fruiting bodies (sporocarps). Fungal sporocarps were identified to species where possible or to genus [53]. We also classified fungal sporocarps as symbiotic (mycorrhizal), saprophytic or pathogenic fungal guild, using the FUNGuild online database [54,55] and updated with databases such as Põlme et al. (2020) [56] and Zanne et al. (2020) [57].

We used two different sampling procedures in each sampling site (Figure 2): (1) transects to quantify the intensity of rooting on a large scale and (2) plots to study the relationship between the abundance of fungal sporocarps (by the fungal guild) and wild boar rooting intensity at the sampling plot level. Thus, we obtained an overall estimate of the rooting extension per site with the transect method. In contrast, during the plot sampling, we obtained a local estimate of the disturbance extension per plot and a count of the fungal sporocarp density and diversity. 

(1)We randomly placed a transect in each of the 11 study sites. In these transects, we recorded the rooting intensity per site as a percentage of soil disturbance along the 200 m long transects [11]. The rate of soil rooting was calculated as follows; a fixed bandwidth of 1 m was established and each rooting length was scored within this band [9,15].(2)The plots were square sampling areas of 2 × 2 m, separated by a minimum of 10 m. Ten plots per site were established in areas with wild boar presence, five in disturbed areas (i.e., with rooting) and five in undisturbed (no rooting) areas. Disturbed and undisturbed plots were placed alternately along the transect (see Figure 2). If, after a distance of 10 m from a previous plot, for instance, a disturbed area, we did not find an area free from wild boar disturbance, we continued until we found an undisturbed area. In areas without presence of wild boar, five plots were sampled in a similar fashion, that is, square sampling areas (2 × 2 m) separated by a minimum of 10 m. In each sampled rooting/disturbed plot, we estimated the rooting intensity as the percentage (as to the closest 10%) of the area of soil altered by wild boar. Therefore, we established three wild boar treatments: (1) without wild boar presence; (2) potentially with animals but no signs of rooting; and (3) with animals and rooting signs (Figure 2). Species richness and abundance of each fungal guild were also measured at the plot level. Each 2 × 2 m plot was monitored by recording and photographing the fungal sporocarps.

### 2.3. Abiotic Variables

We used three environmental variables, namely forest type, soil rockiness and soil humidity, known to influence fungal habitat, and thus the composition, diversity and abundance of fungal sporocarps [58,59,60,61,62]: and we used these to investigate how they could influence the abundance of fungi and fungal guilds. 

(A)Forest type. Classified as closed forest (>70% of forest cover), semi-open forest (between 30–70% of forest cover) and open forest (<30% of forest cover) [63]. The degree of canopy openness in a boreal forest directly influences the amount of light the forest floors receive, subsequently affecting plant composition and air and soil humidity, which are critical conditions for fungal communities [45,64,65].(B)Soil rockiness. Either rocky or soft ground cover, i.e., rocky with a rock cover >50%, whereas soft cover with <50% of rock cover. Soil rockiness influences water soil dynamics [66,67] and likely soil temperature, which may directly modify the conditions for fungal communities [45].(C)Soil humidity. Dry with a soil moisture < 50% and wet > 50%. The humidity of soil is related not only to foliar coverage but also to soil texture and climatic conditions, such as the amount of precipitation and wind [61,68]. We estimated the soil moisture based on wetness on the ground, which in most cases was related to nearness to the River Dalälven or other wetter areas such as bogs or fens commonly intermingled in the boreal forest.

### 2.4. Statistical Analyses

We analyzed the effects of the presence of wild boar and the intensity of wild boar rooting on the abundance of fungal sporocarps, for the overall fungal sporocarps and ectomycorrhizal, pathogenic and saprophytic fungal guilds, by performing Poisson GzLMs with a log-link function, at two spatial levels, sampling site and plot (Table 1). We performed two models at the site level, which are directly related the presence/absence of wild boar (model 1a) and the presence/absence of boar rooting areas (model 1b), with site abundance of fungal sporocarps (as the response variable). At the plot level, we evaluated the effect of the presence of wild boar and wild boar rooting, hereafter wild boar factor (three levels: plots in wild boar rooting, undisturbed plots but in sites with wild boar and undisturbed plots in sites without the presence of wild boar), on the abundance of fungal sporocarps (model 2) and the three fungal guilds separately (models 3, 4 and 5). We performed these latter four models (model 2–5), in which the wild boar factor and the three environmental variables (forest type, soil type and humidity) were used as predictors. For comparison to other factors (models 2–5), we used the unique wild boar factor instead of separating it into presence/absence of wild boar and wild boar rooting intensity. To account for all possible effects of the predictors considered, regardless of significance, we used the full models rather than the ‘best model approach’ (which favors precision vs. bias). The significant *p*–value was set at <0.05. Statistical analyses were performed using InfoStat software.

## 3. Results

The percentage of rooted area ranged between 0% on sites without wild boar and 47% on those with the presence of wild boar (mean ± S.E. = 16.4% ± 19%). The fungal sporocarps ranged between 0 and 163 on the studied plots. We counted 1282 fungal sporocarps’, belonging to 29 different genera (482 symbiotic, 626 saprophytic and 184 pathogenic). The genera *Mycena, Cortinarius, Cantharellus* and *Marasmius* were notably abundant (>100 individuals).

At the site level, we found a significant difference between areas with and without wild boars in fungal sporocarp abundance (model 1a; F = 9.26; *p* < 0.01; Figure 3a), where the presence of wild boars increased sporocarp abundance by ≈20%. However, we did not find a significant difference between disturbed and undisturbed areas for total sporocarp abundance (model 1b, F = 1.32; *p* > 0.05, Figure 3b).

At the plot level (model 2), the overall abundance of fungal sporocarps showed a significant difference among the levels of the wild boar factor (F = 8.39; *p* < 0.05; Table 2). After controlling for environmental factors, the results indicated a significantly higher sporocarp abundance on disturbed plots. Yet no significant differences between plots were found in sites where wild boar is absent and in sites where the surveyed plots remain undisturbed (Table 2). There was a significant positive relationship between both humidity and forest type (highest in the semi-open forest) on the overall abundance of fungal sporocarps, which corroborates our second working hypothesis (model 2).

Regarding the abundance of each fungal guild (Table 3): a greater abundance of symbiotic fungi was found (model 3) in the closed forest habitat type with wet soil and without the presence of wild boar (Figure 4). The abundance of saprophytic fungi (model 4) was greater in the semi-open forest with wet soil and on disturbed plots (Figure 4). Finally, the abundance of pathogenic fungi (model 5) had a significant relationship with the semi-open forest and on undisturbed plots in sites with wild boar presence (Table 3). 

## 4. Discussion

We expect that wild boar rooting would modify the abundance of fungal sporocarps and their relative abundance in terms of three functional guilds: ectomycorrhizal, saprophytic and pathogenic. We found that wild boar significantly affected the sporocarp abundance in the boreal forest. Their activities tend to favor saprophytic fungal guilds more abundant in semi-open forests and wet soils. However, symbiotic fungi were more abundant in closed forests and moist soils where wild boar were not present. In contrast, pathogenic fungi were more frequent in semi-open forests (between 30 and 70% coverage). A greater abundance was observed in undisturbed plots in sampling sites with the presence of wild boars. 

Other studies show that digging mammals, such as rodents, fossorial mammals and others can create suitable sites for fungal growth [69], while vertebrate foraging activities, such as digging, can alter inoculum distribution [70]. In addition, digging acted as traps for organic matter and sites for forming new soil, which had higher fertility and moisture content and lower hardness than undisturbed topsoil [71]. However, while these fossorial mammals dig, the wild boar lifts and erode the soil, i.e., it uses its snout to pry up and pick up chunks of soil instead of “scratching” them. Therefore, this mechanical procedure by the wild boar can cause different effects on the properties of the soil, the seed bank and on the fungi (break mycorrhizal hyphae more), as well as both mixing and oxygenating the soil, accelerating mineralization. It seems this facilitates fungi growth in general [72], although the effect on fungal guilds and its functional repercussions on the ecosystem is unknown. We found a significant positive association between rooting and the overall abundance of fungal sporocarps, particularly on saprophytic sporocarps. Free-living saprotrophic fungi contribute to the decomposition of soil organic matter and nutrient mobilization. These fungi are perhaps more capable of non-homeostatic behavior (the ability to maintain a stable internal condition) than the symbiotic guild, as the former has certain physiological adjustments that allow them to store available nutrients [73,74]. Indeed, the fungal N concentrations of the saprophytic ecological guild are greater than those of symbiotic species, showing a positive relationship between fungal N% and soil N content [75]. Previous studies have shown an increase in the amount of N in soils and vegetation in high-density wild boar areas [76]. In addition, it has been found that nitrate concentrations were larger in rooted areas, thus suggesting alterations in nitrogen transformation processes by boar activities [14,77]. Thus, the disturbance effect of wild boar could increase the abundance of saprophytic fungal sporocarps while disfavoring the symbiotic guild [74]. The symbiotic guild is strongly constrained (near constant C: N and C: P) because mutualistically, any ‘excess’ N or P that is not essential for fungal metabolism would be passed on to the host to maximize fitness (*sensu* optimal foraging theory) [78]. In addition, rooting activity reduces vegetative cover and leaf litter [77], interfering with this guild’s symbiotic relationships. Mycorrhizal symbiotic fungi are also quite sensitive to soil disturbance [31,79,80], as it can directly disrupt the mycorrhizal networks connecting plants and mycorrhizal fungi in the community. However, in this study, we acknowledge that a causal relationship between wild boar rooting and sporocarp abundance cannot be demonstrated, and thus interpretations should be taken with caution. Indeed, we cannot rule out that the wild boar could choose to forage in areas with higher sporocarp abundance. However, after a first soil disturbance, plant and soil responses with an increase in nutrients (positive feedback) are very likely [81]. 

Our study indicates that the wild boar has an important potential effect on the abundance of fungal sporocarps and that its disturbances can enhance some fungal guilds more than others, as is the case of saprophytic fungi. This asymmetry may enhance some processes more than others (nutrient cycling) and modify ecosystem processes and dynamics, thus enhancing a faster nutrient cycling that ultimately impacts the ecological communities. However, our experimental design does not allow us to infer causal relationships between wild boar rooting and sporocarp abundance. Therefore, the observed associations could also be interpreted as a wild boar feeding strategy, selecting those areas with a greater abundance of sporocarps. Consequently, it would be helpful to design studies with enclosures (e.g., fencing) to keep wild boar in and out of plots for detailed fungal studies. Although our results suggest a potential role of wild boar in modulating fungal communities and are coherent with previous pieces of evidence, further studies are required to (1) directly establish the effects of rooting on fungal dynamics, since rooting could affect fungal dynamics by disturbing the soil, increasing root mortality and altering reproductive success (via sporocarps), (2) investigate how wild boar can facilitate the growth of certain fungal species of fungal guilds in disturbed areas and (3) how this, in turn, could affect the plant community assembly and composition and their dynamics.

## 5. Conclusions

Analyzing soil fungi, particularly their fungal guilds, in relation to soil disturbances can help understand the recovery processes after a disturbance. Indeed, a complete understanding of the fungal guilds at the different stages of recovery/succession in a forest can improve our ability to decide on the most suitable fungal community, along with the appropriate forest tree for each reforestation action [82]. For example, our study and Rodriguez-Ramos et al. (2020) [25], found that saprotrophs seem to be followed by mycorrhizal fungi in succession after different disturbances in the boreal forest. This is promising information for future research aimed at the mechanistic understanding of the succession of different fungal guilds and their activities to enhance succession/restoration. All in all, our study indicates that wild boar represent a disturbance agent that is relatively novel in these boreal ecosystems (at least in recent history) and, based on sporocarp abundance and the recurrent nature of wild boar rooting [83], could eventually affect future fungal populations, with implications for forest functioning and dynamics.

## Figures and Tables

**Figure 1 animals-12-02521-f001:**
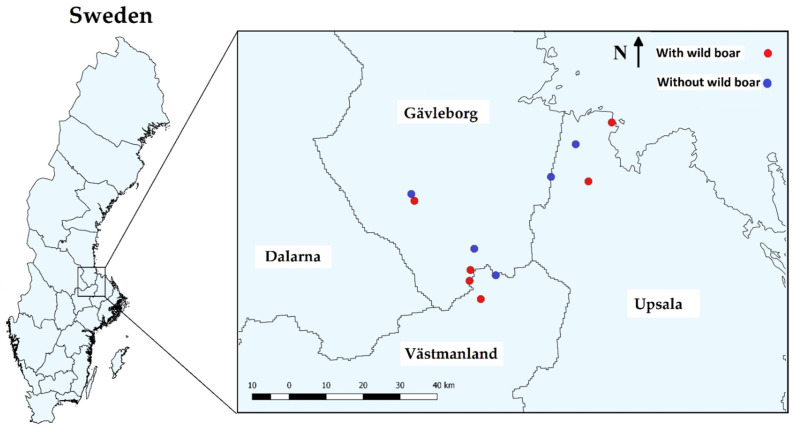
Location of study plots of wild boar-fungal interactions in Gävleborg, Uppsala and Västmanland, central Sweden. Plots with wild boar (red) and without wild boar (blue). Performed using the software QGIS version 2.8.

**Figure 2 animals-12-02521-f002:**
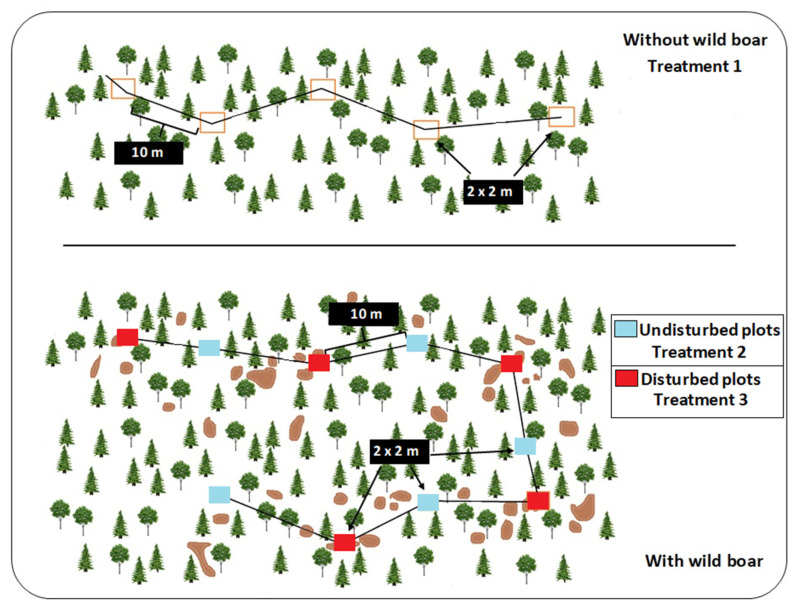
Depiction of the distribution of plots for wild boar (**below**) and without wild boar sites (**above**), used in the wild boar–fungal interactions in Gävleborg, Uppsala and Västmanland, central Sweden.

**Figure 3 animals-12-02521-f003:**
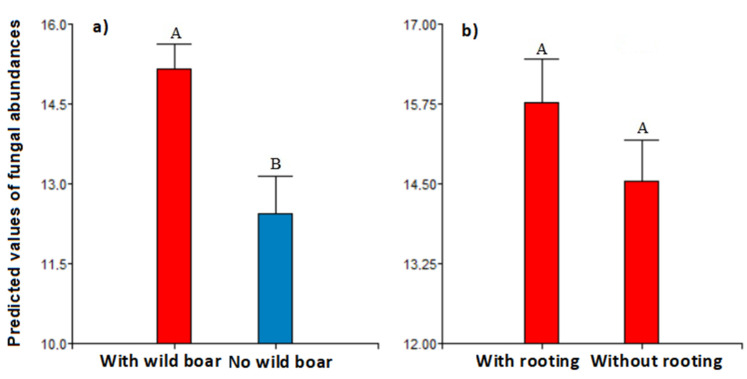
Overall predicted values of the abundance of fungal sporocarps for areas with wild boar and without wild boar (**a**) and disturbed and undisturbed regions (**b**). Bars indicate the standard error. Different letters indicate significant differences among groups according to Fisher’s LSD post-hoc tests (*p* < 0.05).

**Figure 4 animals-12-02521-f004:**
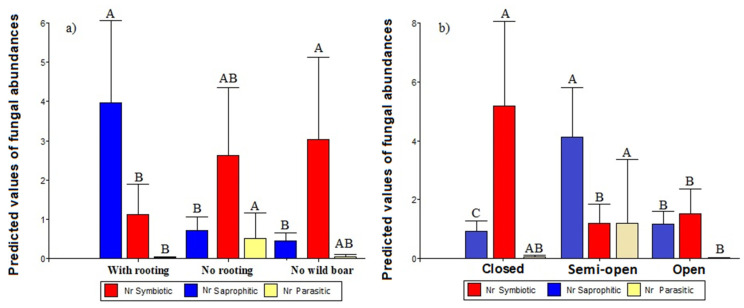
Predicted values of the abundance of fungal sporocarp guilds for each treatment (**a**): wild boar (with rooting), wild boar (no rooting) and no wild boar areas and each forest type (**b**). Bars indicate the standard error. Different letters indicate significant differences among groups according to Fisher’s LSD post-hoc tests (*p* < 0.05). Nr = Number of sporocarps per plots.

**Table 1 animals-12-02521-t001:** List of developed models (at site and plot level).

Site Level		
Models	Predictors	Response variable
Model 1a	Presence/absence wild boar	Abundance of fungal sporocarps
Model 1b	Presence/absence rooting	Abundance of fungal sporocarps
Plot level		
Model 2	Treatment/environmental variables	Abundance of fungal sporocarps
Model 3	Treatment/environmental variables	Abundance of symbiotic
Model 4	Treatment/environmental variables	Abundance of saprophytic
Model 5	Treatment/environmental variables	Abundance of pathogenic

**Table 2 animals-12-02521-t002:** Overall abundance (model 2) of fungal sporocarps, as responses to the presence of wild boar and wild boar rooting (three treatments), analyzed with GzLM. Coefficients for the level of fixed factors were calculated using reference values of the treatment ‘wild boar’ (with rooting) in the variable “treatment”, ‘rocky’ in the variable “type of soil”, ‘open’ in the variable “environmental type” and ‘wet’ in the variable “soil humidity”. (* *p* < 0.05; *** *p* < 0.001).

Variable	*df*	*F*-Value	Coefficient ± E.S.
Abundance of fungi (model 2)			
Intercept	1	19.16 ***	4.14 ± 0.22
Treatment	2	8.39 *	Wild boar no rooting = −1.41 ± 0.14 No wild boar = −1.49 ± 0.29
Type of soil	1	0.18	Soft = 0.09 ± 0.21
Humidity	1	46.42 ***	Dry = −0.88 ± 0.13
Forest type	2	24.05 ***	Semi-open = 0.56 ± 0.13 Closed = 0.33 ± 0.15

df shows the degree of freedom of the numerator.

**Table 3 animals-12-02521-t003:** Fungal guild abundances (model 3, 4 and 5) as responses to the presence of wild boar and wild boar rooting (three treatments) analyzed with GzLMM. Coefficients for the level of fixed factors were calculated using reference values of the treatment ‘wild boar’ (with rooting) in the variable “treatment”, ‘open’ in the variable “forest type”, ‘soft’ in the variable “type of soil”, and ‘wet’ in the variable “humidity”. (* *p* < 0.05; ** *p* < 0.01; *** *p* < 0.001).

Variable	*df*	*F*-Value	Coefficient ± E.S.
Abundance of symbiotic (model 3)			
Treatment	2	5.75 **	Wild boar no rooting = 0.40 ± 0.12 No wild boar = 0.53 ± 0.88
Forest type	2	26.72 ***	Closed = 1.30 ± 0.32 Semi-open = −0.23 ± 0.29
Type of soil	1	3.64	Rocky = −1.29 ± 0.68
Humidity	1	12.15 ***	Dry = −0.93 ± 0.27
Abundance of saprophytic (model 4)			
Treatment	2	25.1 ***	Wild boar no rooting = −0.66 ± 0.09 No wild boar = −1.19 ± 0.68
Forest type	2	42.78 ***	Closed = −0.45 ± 0.21 Semi-open = 1.63 ± 0.19
Type of soil	1	3.71	Rocky = −1.47 ± 0.78
Humidity	1	15.21 **	Dry = −0.70 ± 0.18
Abundance of pathogenic (model 5)			
Treatment	2	3.98 *	Wild boar no rooting = 1.87 ± 0.75 No wild boar = 0.81 ± 1.00
Forest type	2	4.01 *	Closed = 1.19 ± 1.31 Semi-open = 2.80 ± 1.00
Type of soil	1	1.2	Rocky = 0.84 ± 0.77
Humidity	1	3.09	Dry = −1.16 ± 0.66

df shows the degree of freedom of the numerator.

## Data Availability

The data used to support the findings of this study are available from the corresponding author upon request.

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
