# Peer review of "Wild Boar Effects on Fungal Abundance and Guilds from Sporocarp Sampling in a Boreal Forest Ecosystem"

_animals, 2022, doi:10.3390/ani12192521_

Round 1

Reviewer 1 Report

The manuscript entitled "Wild Boar regulating Fungal Guilds in a Boreal Forest Ecosystem" investigated the effects of wild boar on the abundance of mushrooms and their functional guilds. They found that the presence of wild boar and intensity of rooting were associated with the abundance of mushrooms. The manuscript was prepared with care and fully deserves to be published online. Some minor comments are given in the attached PDF.

Author Response

We are very grateful for the reviewer's good opinion of the manuscript. We also appreciate his/her comments that have helped improve the study.

Attached file

Reviewer 2 Report

This is the review for the article by Carpio and colleagues entitled “Wild Boar regulating Fungal Guilds in a Boreal Forest Ecosystem” for publication in the journal “Animals”. The study investigates the impact of wild boar and some additional environmental factors (i.e. soil rockiness, soil wetness and forest type) on the fungal sporocarp abundances in Swedish boreal forest. The authors found that wild boar rooting impact fungal sporocarp abundance, especially the one of saprotrophic fungi. However, the authors correctly conclude that no final statement can come out of their results. Nevertheless, the study is worth to be published after some revisions, since it’s interesting and amongst the first studies on such trophic interactions in forest ecosystems. The article is well introduced and discussed, but the method description and result section lacks clarity at the moment. It’s very appreciable that the authors monitored all those fruiting bodies. In times, where molecular methods are easy and cheap, it’s really important to also continue to do such “classical” approaches. Thank you for this!

Please see my section-wise comments below:

Abstract:

-          Line 23: Please delete “most”.

-          Line 26: “(Sep-Nov…)” and something is wrong with the verb.

-          Line 33: Abundance means number, no?

Introduction:

-          Line 71: Please delete “most”.

-          Line 80: “were”

-          Lines 105-106: But did you really assess the fungal communities? The results display only findings on sporocarp abundance in relation to all of your variables? Please delete or even better add the results on community composition.

-          Line 109-114: This comes a bit unexpected. I believe there would be better possibilities to mention the impact of the environmental factors beforehand in your introduction. And also think of any kind of hypotheses and if it’s simply: “We expect environmental factors to impact fungal abundance.”. But this will sound more confident and professional as “we actually don't know what to expect”. It’s also possible to simply formulate the hypothesis based on your findings. The readership won’t know, when you thought what.

-          Line 115: I guess 1.1. is wrong and this already belongs to the methods.

Methods:

-         I read the methods twice and still I am not entirely sure what you did. I tried to find out what the problem was. I am not entirely certain, but I guess it’s because you repeat information too often. For instance you state that the study was carried out in boreal forest in line 119 (It’s “Norway spruce” by the way.) then comes the map and then again in line 127 you again inform us that’s boreal forest. That the case for several things throughout your explanations. Thus, please check carefully and delete the unnecessary repetitions. I believe, this will sharpen the text and it will be easier to understand.

-          The map: Fig. 1: Please write next to Sweden that this is Sweden. Though it’s in the caption, not everybody knows the shape of Sweden. And the right section has partly a bad quality. The counties are hard to read. Please increase the resolution.

-          Line 127: What’s modern forestry?

-          Line 140: What are professional game keepers?

-          Line 153: You did not perform an experiment, you did a monitoring. Please change! And did I understand right, you had three wild boar treatments: 1) without animals for sure, because some hunters or I guess professional game keepers, know it; 2) potentially with animals but no signs of rooting; and 3) with animals and rooting signs? Could you just state at as simple as possible? And address this also in the figure 2? The resolution of Fig. 2 also needs to be increased.

-          Line 193: Why don’t you call it “forest type”?

-          Line 199: >50% stone? That’s a mountain? Are fungi growing there?

-          Line 209: I also had problems to follow your models and I would suggest a table to list them. And here nothing on fungal communities, though it would be possible, wouldn’t it?

Results:

-          Due to my difficulties to understand the models, I had also problems to follow the result section. Maybe it will come once the models are clearly listed.

-          Fig. 3 and 4: Please clarify on the y-axis that you mean fungal abundances. And I would prefer boxplots or at least error bars in both directions to see the differences. If there is no significant difference, there is no need to indicate it with the same letters. And normally the letters belong over the bars or boxplots (as in your Fig. 4).

Discussion:

-          Line 278: Please delete the sentence on fungal community or add the respective results.

-          Do you saw any signs of wild boars eating the mushrooms somehow? I could imagine that the fruiting bodies of the mycorrhizal fungi are maybe more delicious and thus are not there to find anymore. But I basically agree with you that it’s hard to draw some conclusions since also the production of sporocarp depend on so many things. I would appreciate if you could add a few sentences on molecular methods and how they could potentially overcome this bias.

-          Line 340: There are no fruiting bodies produced by AMF. So what do you mean here?

-          Line 344: Are you sure on “soil fungi”. You did not monitor the fungi on deadwood or other substrates? If so, maybe add a brief comment to the methods.

Author Response

We thank the reviewer for his/her positive feedback and comments that have helped to improve the new version of the manuscript. Now we have improved the description of the methods and results following reviewers’ recommendations.

Attached file

Reviewer 3 Report

The authors present a fairly simple but useful study where the abundance of mushrooms in areas with or without boars were compared.  The contrasting response between saprotrophic and mycorrhizal mushrooms is interesting and underscores the need to consider these animals as novel and potentially important disturbance agents in boreal forests.  My main concern with the interpretations of this work is that the presence of fruiting bodies as an indicator of guild function is pretty tenuous; it is not clear whether the fungal biomass within the soil and the community composition for all guilds has been truly affected by boar rooting.  A useful review on fruiting is Sakamoto 2018. Fungal Biology Reviews 32: 236-248.  For example, a gain in saprotrophic mushrooms might reflect the strategy of certain species to fruit after injury via soil disturbance rather than an increase in fungal dominance.  There may be fewer EMF sporocarps simply because the boars were feeding on them (without disturbing the soil), and again not necessarily reflecting a reduction in symbiotic function.  Overall I would caution the authors that their results indicate an interesting dynamic on fungal reproduction by the presence of wild boars, worthy of more thorough study, but some of their Discussion and Conclusions would require direct measures of soil fungal populations and in situ dynamics.

Title – this title is a stretch as the authors have not delved into how fungal guilds are regulated in any detail, only in the number of sporocarps. Suggest “Wild boar effects on fungal sporocarp abundance in a boreal forest ecosystem”

Line 32. There isn’t any detail really about the diversity of sporocarps, which would generally involve species identification and more comprehensive statistics on community structure.  It would be more accurate to state ‘intensity of rooting and the abundance of sporocarps for three fungal guilds’

Line 38-41. I doubt that a relatively small change in fruiting body abundance from a one year study can be confidently linked to such a bold conclusion on energy transformation and plant diversity. 

Line 83. Wouldn’t direct root mortality from bioturbation be the largest issue regarding boars?  Odd to stress network integrity when the roots are physically removed.  Another form of physical disturbance that might be relevant here is soil compaction as it pertains to fungal vulnerability; see for example Trappe et al. 2009 Can J For Res 39: 1662-1676 and Kranabetter et al. 2017. Forest Ecology and Manage 402: 213-222.

Line 100. As noted above, the number of sporocarps cannot be used to predict ecological consequences, that would take much more detailed study.

Line 106. Was the composition of the fungal community ever tested?  The results look to be entirely devoted to sporocarp abundance

Line 231. Beside range it would be worth noting the average level of rooting disturbance.

Line 232. Should this be ‘counted 1282 fungal sporocarps’?  Identification suggests the authors recorded 1282 separate species

Line 236. Ideally the results would include a comment on effect size rather than solely the p value – e.g. the presence of boars increased sporocarp abundance by 25%

Figure 3. For the y axis, what are the units?  Number of sporocarps per transect?  I’m not sure why these values are predicted, is this the net effect of the boars after correcting for all the other variables?  If so, shouldn’t those factors be tested and specified for the transects (model 1) in a Table as they were for the plots (model 2 etc.)?

Figure 4. What does ‘Nr’ refer to in the Figure caption?  Please explain the units for the Y axis here as well

Line 278. Be careful in some of this terminology; a shift in fungal community composition suggests a more detailed analysis of species abundance.  What the authors have show instead is a shift in fruiting body predominance, with some loss in symbiotic fungi and gains in saprotrophic

Line 319. Odd how much the authors rely on arbuscular studies from greenhouse and agricultural settings.  I would think there is ample work on ectomycorrhizal species from forest settings, why not cite the more relevant literature instead?

Line 336. That is a valid point. Are there any plans to use enclosure studies (e.g. fencing) to keep boars in and out of plots for detailed fungal studies?

Line 338. Might be worth expanding on this point - rooting could affect fungal dynamics by disturbing the soil, increasing root mortality and altering reproductive success (via sporocarps)

Conclusions –Recovery from disturbance, as in the Ramos-Rodriguez study, was after stand mortality which is quite different than the microsite soil disturbances scattered in a mature forest as studied here.  This speculation about saprotrophic replacement by EMF is hard to follow.  Perhaps the more salient point to conclude with is that boars represent a disturbance agent that is presumably novel to these boreal ecosystesm (at least in recent history) and, based on sporocarp abundance, could eventually affect some fungal populations.  If boar populations are high and rooting both extensive and frequent would these animals not also seriously undermine tree health and productivity (just based on root mortality)? 

Author Response

We thank the reviewer for the positive feedback and comments that have helped improve our manuscript. 

Attached file

Round 2

Reviewer 2 Report

Thank you for addressing all my points.

I still believe a link to molecular tools in the discussion or conclusion would make sense.

However, I congratulate you on this nice paper and hope it gets cited very often!

Author Response

Thank you for your good acceptance of the manuscript.

Reviewer 3 Report

Good job with the revisions

Author Response

(The authors gave the same response as above.)
